# PRIMAL OPTIMISM IN ONLINE OPTIMIZATION

## ABSTRACT

We consider the classic online convex optimization problem in which an algorithm outputs vectors $z_t$ in response to vectors $g_t$. Our algorithm seeks to improve the regret when it has access to a sequence of "hint" vectors $v_t$ that estimate the location of the final optimal parameter value $\mathring{u}$. Specifically, we provide an online linear optimization algorithm that guarantees regret $R_T(\mathring{u}) = \sum_{t=1}^{T} \langle g_t, z_t - \mathring{u} \rangle \leq \sqrt{\sum_{t=1}^{T} \|g_t\|^2 \|v_t - \mathring{u}\|^2}$ for any comparison point $\mathring{u}$ and any sequence of vectors $v_t$, so long as $v_t$ is available before we commit to $z_{t+1}$.

## 1 BACKGROUND AND GOALS

Almost all large machine learning models today are trained using first-order stochastic optimization methods. This paper develops new analytical techniques for improving the convergence guarantees of such methods. In particular, we develop a method for incorporating a "guess" for what the optimal model parameters should be. The guess may vary over time as new information is acquired. Our method provides a formal guarantee for improved convergence when the guess is high quality, and is developed using the online linear optimization framework.

Online linear optimization (Shalev-Shwartz, 2007; McMahan, 2014; Orabona, 2019; Hazan, 2019) is a successful paradigm for designing first order stochastic optimization algorithms - for example, this style of analysis was used as motivation for successful optimizers such as Adam (Kingma & Ba, 2014) and Shampoo (Gupta et al., 2018). In this setting, an algorithm outputs a vector $z_t$ in some convex domain $W$, and then sees a "loss vector" (or "gradient") $g_t$. The goal is to control the *regret*, defined as follows:

$$R_T(\mathring{u}) = \sum_{t=1}^{T} \langle g_t, z_t - \mathring{u} \rangle$$

Here $\mathring{u} \in W$ is an arbitrary comparison point[1] This seemingly abstract setting is closely connected to the standard stochastic convex optimization problem: if $\mathbb{E}[g_t] = \nabla F(z_t)$ for some convex loss $F$, then for $\overline{w} = \frac{1}{T} \sum_{t=1}^{T} z_t$, we have:

$$\mathbb{E}[F(\overline{w}) - F(\mathring{u})] \leq \frac{\mathbb{E}[R_T(\mathring{u})]}{T}$$

This method is known as "Polyak-Ruppert averaging" (Polyak, 1990; Ruppert, 1988), or the "online-to-batch conversion" (Cesa-Bianchi et al., 2004).

A common goal in online optimization algorithm design is to obtain various forms of *adaptivity* (Duchi et al., 2010; McMahan & Streeter, 2010; Mcmahan & Streeter, 2012). An algorithm is "adaptive" if it is able to obtain lower regret on "easy" sequences of losses without significantly sacrificing performance on more adversarial loss sequences. One desirable adaptive bound is (Orabona & Pál, 2016; Jacobsen & Cutkosky, 2022; Zhang et al., 2022; Mhammedi & Koolen, 2020)

$$\sum_{t=1}^{T} \langle g_t, z_t - \mathring{u} \rangle \leq \sqrt{\sum_{t=1}^{T} \|g_t\|^2 \|\mathring{u}\|^2} \tag{1}$$

---

[1] A seemingly more general online *convex* optimization setting can be reduced to the linear case, so it suffices to consider only linear regret as done here Zinkevich (2003).

The above bound ensures that the regret is small when the gradients or the comparison point $\mathring{u}$ happen to be small.

Some algorithms can go even further: given a sequence of "hints" $h_1, \ldots, h_T$ for the values of $g_1, \ldots, g_T$, these algorithms obtain regret of the form (Rakhlin & Sridharan, 2013):

$$\sum_{t=1}^{T} \langle g_t, z_t - \mathring{u} \rangle \leq \sqrt{\sum_{t=1}^{T} \|g_t - h_t\|^2 \|\mathring{u}\|^2}$$

Such algorithms are frequently called "optimistic" because the regret bound improves when the hints are very informative. We will call this type of guarantee "dual optimism" because the hints $h_t$ are providing information about the values of $g_t$, which live in the dual space.

Optimistic bounds allow the regret (and hence convergence rate for stochastic optimization) to improve when the problem is in some way "non-worst-case", or in general displays certain kinds of predictible patterns in the losses. For example, for smooth losses, we can use the smoothness property to successfully predict that the gradients $g_t$ will change only very slowly: setting $h_t = g_{t-1}$. This enables us to recover accelerated gradient descent bounds (Joulani et al., 2020).

### 1.1 Our Goal: Primal Optimism

In contrast to dual optimism, our goal is to build an algorithm that takes advantage of hints $v_t \in W$, which are estimates of $\mathring{u}$ rather than $g_t$:

$$\sum_{t=1}^{T} \langle g_t, w_t - \mathring{u} \rangle \leq \sqrt{\sum_{t=1}^{T} \|g_t\|^2 \|\mathring{u} - v_t\|^2} \tag{2}$$

Since the hints $v_t$ are attempting to providing information about a primal variable, $\mathring{u}$, we call this "primal optimism'. Intuitively, we might expect it to be easier to form accurate "primal" hints $v_t$ rather than "dual" hints $h_t$ because, unlike $g_t$, $\mathring{u}$ does not change over time and is not affected by noise in the gradients.

Although we not aware of prior work establishing this general bound, certain special cases do exist. The algorithms of van Erven & Koolen (2016); Cutkosky & Orabona (2018); Wang et al. (2020) provide similar results for the particular setting $v_t = z_t$. We generalize these results to obtain a primal optimistic algorithm for any sequence $v_t$. We also avoid a logarithmic memory overhead incurred by van Erven & Koolen (2016); Wang et al. (2020). Similar to dual optimism, we allow our sequences $v_t$ to be generated during the run of the algorithm, even allowing $v_t$ to depend on $z_t$.

Importantly, our method is a *reduction*: given any sufficiently capable online optimization algorithm, we transform it into one that achieves the primal optimism bound (2). This means that our method can be easily integrated into both current and future optimization analyses with minimal struggle.

### 1.2 Primal Optimism as "Soft Restarting"

Let's take a moment to gain some intuition for what this "primal optimism" bound might be doing. A common paradigm in optimizer analysis is to view the initialization point of the optimizer as indicating some kind of "prior" for the optimal parameter value. Mathematically, this intuition manifests in convergence bounds that improve in some way if the initialization point is close to the optimal parameters. For example, the standard suboptimality bound for stochastic gradient descent is $\frac{G\|z_1 - \mathring{u}\|}{\sqrt{T}}$, and so improves when the initialization point $z_1$ is close to the optimal parameters $\mathring{u}$. This means that one can frequently improve the performance of an optimization algorithm by "restarting" it periodically: after running for a certain number of steps, we have a better idea for the optimal parameter $\mathring{u}$, so we restart with $z_1$ equal to this guess. This is a classic technique in the optimization toolbox (Nesterov, 2013).

With this background, we can interpret the "primal optimism" bound as providing a smoothed-out version of the restarting procedure: instead of restarting the algorithm every $K$ iterates using a new guess for $\mathring{u}$, we allow ourselves to update the guess on every single iteration (if desired) without a full restart.

### 1.3 PAPER ORGANIZATION

in Section 1.4 we define the online optimization setting provide some common notation to be used throughout this paper. In Section 2 we provide a further motivating application of primal optimism, demonstrating an area in which this bound concretely improves convergence behavior. In Section 3 we provide a some high-level intuition for how our analysis will proceed. In Section 4 we provide a proof of our main results. Finally, in Section 5 we interpret our algorithm in the context of momentum methods and provide some remarks on future directions.

### 1.4 SETTING AND NOTATION

We use $H$ to denote a $d$-dimensional real vector space. We use $\|\cdot\|$ to denote the usual 2-norm on $H$. $W \subset H$ is a convex domain with finite diameter $D = \sup_{x,y \in W} \|x - y\|$.

In the online linear optimization game, for each $t \in \{1, \ldots, T\}$, an algorithm commits to a vector $z_t \in W$, and then observes a vector $g_t \in H^\star$. After all $T$ rounds, we observe a comparison point $\mathring{u} \in W$ and measure the regret $R_T(\mathring{u})$. We will assume throughout this paper that $\|g_t\| \leq G$ for all $t$ for some fixed $G$, but otherwise make no assumptions about $g_t$ or $\mathring{u}$. We will allow our algorithms to also observe a sequence of "hints" $v_t \in W$ where $v_t$ is revealed prior to choosing $z_t$. No further assumptions are made about $v_t$.

We will use $\tilde{O}$ notation to hide constants logarithmic factors in $\|g_t\|$, $T$, $D$, and $G$. We use the compressed-index notation $\alpha_{a:b}$ to indicate the sum $\sum_{i=a}^b \alpha_i$, and similarly write $\|g\|_{a:b}^2 = \sum_{i=a}^b \|g_i\|^2$.

Finally, our method will make use of a "base" online optimization algorithm $\mathcal{A}$ with domain $H$ rather than $W$. We will construct our final online algorithm by calling $\mathcal{A}$ as a black-box subroutine (i.e, our result is a reduction). We use $\hat{z}_t \in H$ to indicate the outputs of $\mathcal{A}$ and $\hat{g}_t$ to indicate the input losses to $\mathcal{A}$. The regret of $\mathcal{A}$ is denoted by $R_t^{\mathcal{A}}$:

$$R_t^{\mathcal{A}}(\mathring{u}) = \sum_{i=1}^t \langle \hat{g}_i, \hat{z}_i - \mathring{u} \rangle$$

## 2 APPLICATION: ADAPTIVITY TO STRONG-CONVEXITY IN STOCHASTIC CONVEX OPTIMIZATION

Recent work (Cutkosky, 2019; Kavis et al., 2019; Defazio et al., 2023; 2024) have developed advanced online-to-batch conversion results that yield empirically successful optimization algorithms. The most general of these is that of Defazio et al. (2024), which converts any online optimizer into a stochastic convex optimization algorithm with the guarantee:

$$\mathbb{E}[F(x_T) - F(\mathring{u})] \leq \frac{\mathbb{E}[\sum_{t=1}^T \langle g_t, z_t - \mathring{u} \rangle - B_F(\mathring{u}, y_t)]}{T}$$

Here $F$ is a convex objective, $y_t$ are carefully selected points that depend on $z_t$ at which stochastic gradients $g_t$ satisfying $\mathbb{E}[g_t] \in \partial F(y_t)$ are evaluated, $B_F$ is the Bregman divergence, and $x_T$ is also a carefully selected final output. Importantly, unlike the classical online-to-batch/Polyak-Ruppert averaging approach, the gradients are evaluated at $y_t \neq z_t$.

Now, suppose that $F$ is $\mu$-strongly convex. This implies $B_F(\mathring{u}, y_t) \geq \frac{\mu}{2} \|\mathring{u} - y_t\|^2$. Then, let us set $v_t = y_t$, suppose $\|g_t\| \leq G$, and suppose our online learner achieves a "primal optimistic" regret bound like (2). Then we have:

$$\begin{aligned}
\mathbb{E}[F(x_T) - F(\mathring{u})] &\leq \frac{\mathbb{E}\left[G\sqrt{\sum_{t=1}^T \|y_t - \mathring{u}\|^2} - \frac{\mu}{2}\sum_{t=1}^T \|y_t - \mathring{u}\|^2\right]}{T} \\
&\leq \sup_Q \frac{G\sqrt{Q} - \frac{\mu Q}{2}}{T} \\
&\leq \frac{G^2}{2T\mu}
\end{aligned} \tag{3}$$

so that this overall bound automatically achieves the correct strongly-convex rate without any explicit knowledge of the strong-convexity constant. Notice the method also gracefully degrades to $O(1/\sqrt{T})$ when $\mu = 0$.

The ability to choose arbitrary $v_t$ was critical here: if we are forced to set $v_t = z_t$ (as is the case with previous primal optimism-style results), then we can only achieve this strong-convexity adaptivity for the standard online-to-batch conversion.

As a concrete example of this application, Defazio et al. (2023) studies the "linear decay schedule":

$$\Delta_t = z_{t+1} - z_t$$
$$y_{t+1} = y_t + \frac{T-t}{T}\Delta_t$$
$$x_T = y_T$$

The bound (3) holds for these values. That is, if we consider $y_1, \ldots, y_T$ to be the "true" iterates of our algorithm (where the gradient are evaluated), then we obtain a so-called last-iterate guarantee. This ensures optimal convergence for convex objectives, but may be suboptimal for strongly-convex objectives. Fortunately, if the $z_t$ are generated by our algorithm with $v_t = y_t$, we will obtain an optimal last-iterate guarantee for strongly-convex objectives.

## 3 SKETCH OF APPROACH

We achieve primal optimism by reducing the problem to online linear optimization over the entire vector space $H$ rather than the finite-diameter domain $D$. Specifically, we take any algorithm $\mathcal{A}$ that outputs (unconstrained) iterates $\hat{z}_t$ in response to gradients $\hat{g}_t$ and obtains regret

$$R_T^{\mathcal{A}} := \sum_{t=1}^{T} \langle \hat{g}_t, \hat{z}_t - \mathring{u} \rangle \leq \tilde{O}\left( \|\mathring{u}\| \sqrt{\sum_{t=1}^{T} \|\hat{g}_t\|^2} \right) \tag{4}$$

in the unconstrained setting in which $\hat{z}_t$ and $\mathring{u}$ can be any elements of $H$, and convert it into one that outputs iterates $z_t \in W$ and obtains regret

$$\sum_{t=1}^{T} \langle g_t, z_t - \mathring{u} \rangle \leq \tilde{O}\left( D \max_t \|g_t\| + \sqrt{\sum_{t=1}^{T} \|g_t\|^2 \|\mathring{u} - v_t\|^2} \right)$$

for any bounded domain $W$ such that $W$ has diameter at most $D$. Note that there are many algorithms achieving (4) in the literature (Kempka et al., 2019; van der Hoeven, 2019; Mhammedi & Koolen, 2020; Jacobsen & Cutkosky, 2022; Zhang et al., 2024) and so it is straightforward to instantiate this reduction.

We accomplish this using two steps. First, we use a reduction from constrained optimization to unconstrained optimization provided by previous work (Cutkosky, 2020), which will allow us to work in a constraint set rather than the entire vector space. Second (and more importantly), we improve and generalize the "offset-by-average" technique introduced by Cutkosky & Boahen (2017); Cutkosky & Orabona (2018) which has been previously used to obtain primal optimism in the special case that $v_t = z_t$. Note that we use this technique instead of the method of van Erven & Koolen (2016); Wang et al. (2020) because the latter method requires a grid of $\log(T)$ expert algorithms that incurs a significant computational and memory overhead. That said, the logarithmic factors in our regret bound will be worse, so it may be that the analysis can be improved.

### 3.1 INTUITION FOR OVERALL TECHNIQUE

Following Cutkosky & Boahen (2017), consider the random variable $V$ that takes value $v_t$ with probability proportional to $\|g_t\|^2$. Then by the bias-variance decomposition, we have:

$$\sum_{t=1}^{T} \|g_t\|^2 \|v_t - \mathring{u}\|^2 = \sum_{t=1}^{T} \|g_t\|^2 \|\mathring{u} - \mathbb{E}[V]\|^2 + \sum_{t=1}^{T} \|g_t\|^2 \|v_t - \mathbb{E}[V]\|^2$$

Now consider running an online algorithm $\mathcal{A}$ that outputs iterate $\hat{z}_t$ and obtains the bound (4), but instead of using the iterates $\hat{z}_t$ directly, we instead output $z_t = \hat{z}_t + \mathbb{E}[V]$. This has the effect of shifting coordinates so that the origin is now at $\mathbb{E}[V]$ and so we would obtain a bound of the form $R_T^{\mathcal{A}}(\mathring{u} - \mathbb{E}[V]) = \sqrt{\sum_{t=1}^T \|g_t\|^2 \|\mathring{u} - \mathbb{E}[V]\|^2}$, which is the bias part of the above equation. Since we do not know $\mathbb{E}[V]$ ahead of time, we approximate it with the instantaneous average $\overline{v}_{t-1} = \frac{\sum_{i=1}^{t-1} \|g_i\|^2 v_i}{\sum_{i=1}^{t-1} \|g_t\|^2}$, so that we play the vector $\hat{z}_t + \overline{v}_{t-1}$ on the $t$th iteration. Bounding how well this approximation reflects $\mathbb{E}[V]$ will yield the variance part of the above equation. While this high-level idea is not new, we generalize and somewhat simplify the analysis to allow for arbitrary $v_t$ rather than the specific case of $v_t = z_t$ considered previously.

Almost all of the difficulty in applying this offset technique lies in bounding the effect of using the running averages $\overline{v}_{t-1}$ as offsets rather than the true average $\mathbb{E}[V] = \overline{v}_T$. To do this, we will need to make use of the fact that $W$ is bounded. This actually introduces an issue: how can we ensure that $z_t + \overline{v}_{t-1} \in W$ properly? Let's ignore this for now, and assume for purposes of exposition that both the $\hat{z}_t$ values produced by $\mathcal{A}$ as well as the offset values $\hat{z}_t + \overline{v}_{t-1}$ are both in $W$. Then we can write the linearized regret as:

$$\sum_{t=1}^T \langle g_t, \hat{z}_t + \overline{v}_{t-1} - \mathring{u} \rangle$$

$$= \underbrace{\sum_{t=1}^T \langle g_t, \hat{z}_t - (\mathring{u} - \overline{v}_T) \rangle}_{R_T^{\mathcal{A}}(\mathring{u} - \overline{v}_T)} + \underbrace{\sum_{t=1}^T \langle g_t, \overline{v}_{t-1} - \overline{v}_T \rangle}_{\text{error to control}}$$

The first term on the RHS can be bounded appropriately using the regret bound for $\hat{z}$, and yields a term similar to the "bias" term in the bias-variance decomposition. The second term is more difficult, but by exploiting the fact that $\overline{v}_t$ moves very slowly, we can bound this term by roughly the "variance" term in the bias-variance decomposition.

## 3.2 Enforcing the Constraint

In order to deal with the constraints issue mentioned previously, we employ a previously developed reduction Cutkosky & Orabona (2018); Cutkosky (2020). The approach is simple: let $\hat{z}_t^v = \hat{z}_t + \overline{v}_{t-1}$ be the unconstrained iterate, and let $z_t = \Pi_W \hat{z}_t^v$ be the projection to the constraint set. Then we play $z_t$ and get a gradient $g_t$. We post-process this gradient as follows:

$$\hat{g}_t = g_t - \frac{(z_t - \hat{z}_t^v)[\langle g_t, z_t - \hat{z}_t^v \rangle]_+}{\|z_t - \hat{z}_t^v\|^2}$$

where $[x]_+ = \max(0, x)$. The idea is that if $\hat{z}_t^v \in W$, we do not change $g_t$. Otherwise, if $g_t$ is such that $-g_t$ is pointing "into" the constraint, then we still do not channge $z_t$. Finally, if $g_t$ is pointing "out" of the constraint, then we remove the component of $g_t$ that is normal to the constraint set (so that now $\hat{g}_t$ will tangent to $W$).

Formally, Cutkosky (2020) shows the following identity for any $\mathring{u} \in W$:

$$\langle g_t, z_t - \mathring{u} \rangle = \langle \hat{g}_t, \hat{z}_t^v - \mathring{u} \rangle \tag{5}$$

This implies that to bound the regret in the constrained setting, it suffices to send the post-processed $\hat{g}_t$ to the unconstrained algorithm, which outputs $\hat{z}_t^v = \hat{z} + \overline{v}_{t-1}$, which are then projected back to $z_t \in W$. Notice also that by construction, $\|\hat{g}_t\| \leq \|g_t\|$.

## 4 Algorithm and Analysis

Our overall algorithm implementing the intuition presented in Section 3 is presented in Algorithm 1. It achieves the regret bound described by Theorem 2. Theorem 2 is very general result, so we first state an important corollary that shows that Algorithm 1 indeed achieves the primal optimism guarantee.

---

**Algorithm 1** Primal Optimism

---

**Input:** online optimization algorithm $\mathcal{A}$, initial point $x_0 \in W$, initial weight $\alpha_0$, initial vector $v_9$

Set $\overline{v}_0 = v_0$.
**for** $t = 1 \ldots T$ **do**
    Get $\hat{z}_t \in H$ from $\mathcal{A}$.
    Set $\hat{z}_t^v = \hat{z}_t + \overline{v}_{t-1}$
    # Perform "constraint set reduction".
    Set $z_t = \Pi_W(\hat{z}^v)$.
    Play $z_t$, get $g_t$.
    Set $\hat{g}_t = g_t - \frac{(z_t - \hat{z}_t^v)[\langle g_t, z_t - \hat{z}_t^v \rangle]_+}{\|z_t - \hat{z}_t^v\|^2}$ # remove normal component of $g_t$ if necessary.
    Send $\hat{g}_t$ to $\mathcal{A}$ as $t$th gradient.
    Set $\alpha_t = \|\hat{g}_t\|^2$.
    Get $v_t$, and set $\overline{v}_t = \frac{\sum_{i=0}^t \alpha_i v_i}{\alpha_{0:t}}$.
**end for**

---

**Corollary 1.** *If $\mathcal{A}$ guarantees $R_t^{\mathcal{A}}(\mathring{u}) \leq \tilde{O}\left(\|u\|\sqrt{\sum_{t=1}^T \|\hat{g}_t\|^2}\right)$ (e.g. as described in Section 3), then Algorithm 1 ensures:*

$$R_T(\mathring{u}) \leq \tilde{O}\left(\sqrt{\alpha_0\|v_0 - \mathring{u}\|^2 + \sum_{t=1}^T \|g_t\|^2\|v_t - \mathring{u}\|^2}\right)$$

*Proof.* Notice that the guarantee of $\mathcal{A}$ means that we can take $\psi_t(x)$ and $\phi_t(x)$ in Theorem 2 to be $\tilde{O}(1)$, and so the conclusion follows. $\qquad\square$

**Theorem 2.** *Suppose $\mathcal{A}$ guarantees regret $R_t^{\mathcal{A}}(u) = \sum_{i=1}^t \langle \hat{g}_i, \hat{z}_i - u \rangle \leq \epsilon + \psi_t(\|u\|)\|u\| + \phi_t(\|u\|)\|u\|\sqrt{\alpha_0 + \sum_{i=1}^t \|\hat{g}_i\|^2}$ for all $u$ and $t$, where $\epsilon > 0$, and $\psi_t(x)$ and $\phi_t(x)$ are increasing functiosn of $t$ and $x$. Then, Algorithm 1 guarantees both $\|\hat{g}_t\| \leq \|g_t\|$ for all $t$ and also:*

$$\sum_{t=1}^T \langle g_t, z_t - \mathring{u} \rangle \leq \epsilon + \|v_T - \mathring{u}\|\psi_T(\|v_T - \mathring{u}\|) + \left[\epsilon + D\psi_T\left(D + 2D\sqrt{T}\right)\right]\log(1 + \|\hat{g}\|_{1:T}^2/\alpha_0)$$

$$+ \left[1 + \phi_T\left(D + 2D\sqrt{T}\right)\right]\sqrt{\log(1 + \|\hat{g}\|_{1:T}^2/\alpha_0)}\sqrt{D^2(\alpha_0 + \max_t \|\hat{g}_t\|^2 \log(1 + \|\hat{g}\|_{1:T}^2/\alpha_0))}$$

$$+ \left[1 + 2\phi_T\left(D + 2D\sqrt{T}\right)\right]\sqrt{\alpha_0\|v_0 - \mathring{u}\|^2 + \sum_{t=1}^T \|\hat{g}_t\|^2\|v_t - \mathring{u}\|^2}$$

*Proof.* Define $\alpha_t = \|\hat{g}_t\|^2$ for $t \geq 1$, as in Algorithm 1. Then following Cutkosky & Orabona (2018); Cutkosky & Boahen (2017), we consider for purposes of intuition a random variable $V$ that takes on value $v_t$ with probability proportional to $\alpha_t$. We define:

$$\mathbb{E}[V] = \frac{\sum_{t=0}^T \alpha_t v_t}{\sum_{t=0}^T \alpha_t} = \overline{v}_T$$

$$\sum_{t=0}^T \alpha_t \mathbb{E}[\|V - \mathring{u}\|^2] = \sum_{t=0}^T \alpha_t\|v_t - \mathring{u}\|^2$$

and by bias-variance decomposition we can further write

$$\sum_{t=1}^T \alpha_t(\mathbb{E}[\|V - \mathbb{E}[V]\|^2] + \|\mathbb{E}[V] - \mathring{u}\|^2) = \sum_{t=1}^T \alpha_t\|v_t - \mathring{u}\|^2$$

so that we can rephrase our objective as (ignoring constants and logarithmic terms):

$$R_T(\mathring{u}) \le \epsilon + 2\sqrt{\sum_{t=0}^{T} \alpha_t (\mathbb{E}[\|V - \mathbb{E}[V]\|^2] + \|\mathbb{E}[V] - \mathring{u}\|^2)}$$

Since $\sqrt{a} + \sqrt{b} \le 2\sqrt{a+b}$, it suffices to show

$$R_T(\mathring{u}) \le \epsilon + \sqrt{\sum_{t=0}^{T} \alpha_t \|\overline{v}_T - \mathring{u}\|^2} + \sqrt{\sum_{t=0}^{T} \alpha_t \|v_t - \overline{v}_T\|^2]}$$

Now let us examine the regret:

$$R_T(\mathring{u}) = \sum_{t=1}^{T} \langle g_t, z_t - \mathring{u} \rangle$$

using (5):

$$\le \sum_{t=1}^{T} \langle \hat{g}_t, \hat{z}_t^v - \mathring{u} \rangle$$

$$= \sum_{t=1}^{T} \langle \hat{g}_t, \hat{z}_t - (\mathring{u} - \overline{v}_T) \rangle + \sum_{t=1}^{T} \langle \hat{g}_t, \overline{v}_{t-1} - \overline{v}_T \rangle$$

$$= R_T^{\mathcal{A}}(\mathring{u} - \overline{v}_T) + \sum_{t=1}^{T} \langle \hat{g}_t, \overline{v}_{t-1} - \overline{v}_T \rangle$$

$$\le \epsilon + \|\overline{v}_T - \mathring{u}\| \psi_T(\|\overline{v}_T - \mathring{u}\|)$$

$$+ \phi_T(\|\overline{v}_T - \mathring{u}\|)\sqrt{\alpha_0 \|v_T - \mathring{u}\|^2 + \sum_{t=1}^{T} \|\hat{g}_t\|^2 \|\overline{v}_T - \mathring{u}\|^2} + \sum_{t=1}^{T} \langle \hat{g}_t, \overline{v}_{t-1} - \overline{v}_T \rangle \quad (6)$$

The first-few terms in the above expression roughly match the "bias" part of the bias-variance decomposition. So, it remains to prove that $\sum_{t=1}^{T} \langle \hat{g}_t, \overline{v}_{t-1} - \overline{v}_T \rangle$ matches the "variance" part.

Let us start by calculating an alternative form for this sum:

$$\sum_{t=1}^{T} \langle \hat{g}_t, \overline{v}_{t-1} - \overline{v}_T \rangle - \sum_{t=1}^{T-1} \langle \hat{g}_t, \overline{v}_{t-1} - \overline{v}_{T-1} \rangle = \left\langle \sum_{t=1}^{T} \hat{g}_t, \overline{v}_{T-1} - \overline{v}_T \right\rangle$$

After telescoping:

$$\sum_{t=1}^{T} \langle \hat{g}_t, \overline{v}_{t-1} - \overline{v}_T \rangle = \sum_{t=1}^{T} \langle \hat{g}_{1:t}, \overline{v}_{t-1} - \overline{v}_t \rangle$$

Now we apply Lemma 3 to see that for any $K_t$ to be chosen later:

$$\langle \hat{g}_{1:t}, \overline{v}_{t-1} - \overline{v}_t \rangle \le \left[ \frac{R_t^{\mathcal{A}}\left(K \frac{\overline{v}_{t-1} - \overline{v}_t}{\|\overline{v}_{t-1} - \overline{v}_t\|}\right)}{K_t} + \frac{D \sum_{i=1}^{t} \|\hat{g}_i\|}{K_t} \right] \|\overline{v}_{t-1} - \overline{v}_t\|$$

Use the assumed form of $R_t^{\mathcal{A}}$ to obtain:

$$\langle \hat{g}_{1:t}, \overline{v}_{t-1} - \overline{v}_t \rangle \le \left[ \frac{\epsilon + D \sum_{i=1}^{t} \|\hat{g}_i\|}{K_t} + \psi_t(K_t) + \phi_t(K_t)\sqrt{\sum_{i=1}^{t} \|\hat{g}_i\|^2} \right] \|\overline{v}_{t-1} - \overline{v}_t\|$$

Next, we examine $\overline{v}_{t-1} - \overline{v}_t$:

$$\alpha_{0:t}\overline{v}_t - \alpha_{0:t-1}\overline{v}_{t-1} = \alpha_t v_t$$

$$\overline{v}_t - \overline{v}_{t-1} = \frac{\alpha_t(v_t - \overline{v}_{t-1})}{\alpha_{0:t}}$$

So that

$$\langle \hat{g}_{1:t}, \overline{v}_{t-1} - \overline{v}_t \rangle \le \frac{(\epsilon + D\sum_{i=1}^t \|\hat{g}_i\|)\|v_t - \overline{v}_{t-1}\|\alpha_t}{K_t \alpha_{0:t}} + \psi_t(K_t)\frac{\alpha_t\|v_t - \overline{v}_{t-1}\|}{\alpha_{0:t}} + \phi_t(K_t)\frac{\alpha_t\|v_t - \overline{v}_{t-1}\|}{\sqrt{\alpha_{0:t}}}$$

Now, we set $K_t$ to be:

$$K_t = \frac{\epsilon + \sum_{i=1}^t \|\hat{g}_i\|D}{\epsilon/D + \sqrt{\alpha_{0:t}}} \le D(1 + 2\sqrt{T})$$

The inequality follows since $\sum_{i=1}^t \|\hat{g}_i\| \le \sqrt{T}\sqrt{\sum_{i=1}^t \|\hat{g}_i\|^2} \le \sqrt{T}\sqrt{\alpha_{0:t}}$.

With this value, we have:

$$\langle \hat{g}_{1:t}, \overline{v}_{t-1} - \overline{v}_t \rangle \le \epsilon \cdot \frac{\alpha_t}{\alpha_{0:t}} + \psi_t(K_t)\frac{\alpha_t\|v_t - \overline{v}_{t-1}\|}{\alpha_{0:t}} + (1 + \phi_t(K_t))\frac{\alpha_t\|v_t - \overline{v}_{t-1}\|}{\sqrt{\alpha_{0:t}}}$$

Let's define $K = \max_t K_t \le D(1 + 2\sqrt{T})$. Then, since $\psi_t$ and $\phi_t$ are increasing in $t$, we have:

$$\sum_{t=1}^T \langle \hat{g}_{1:t}, \overline{v}_{t-1} - \overline{v}_t \rangle \le (\epsilon + D\psi_T(K))\log(\alpha_{0:T}/\alpha_0) + (1 + \phi_T(K))\sqrt{\log(\alpha_{0:T}/\alpha_0)}\sqrt{\sum_{t=1}^T \alpha_t\|v_t - \overline{v}_{t-1}\|^2}$$

Now observe that $\overline{v}_{t-1}$ is actually the output of the Follow-the-Regularized-Leader algorithm on the losses $f_t(\overline{v}) = \alpha_t\|v_t - \overline{v}\|^2$ with constant regularizer $\alpha_0\|v_0 - \overline{v}\|^2$. Therefore by standard analysis of this algorithm (e.g. see McMahan (2014)), we have $\sum_{t=1}^T \alpha_t\|v_t - \overline{v}_{t-1}\|^2 - \alpha_t\|v_t - \overline{v}_T\|^2 \le D^2(\alpha_0 + \max_{t\ge 1} \alpha_t \log(\alpha_{0:T}/\alpha_0))$, so that we have

$$\sum_{t=1}^T \langle \hat{g}_t, \overline{v}_{t-1} - \overline{v}_T \rangle \le (\epsilon + D\psi_T(K))\log(\alpha_{0:T}/\alpha_0)$$

$$+ (1 + \phi_T(K))\sqrt{\log(\alpha_{0:T}/\alpha_0)}\sqrt{D^2(\alpha_0 + \max_t \alpha_t \log(\alpha_{0:T}/\alpha_0)) + \sum_{t=1}^T \alpha_t\|v_t - \overline{v}_T\|^2}$$

$$\le \left[\epsilon + D\psi_T\left(D + 2D\sqrt{T}\right)\right]\log(\alpha_{0:T}/\alpha_0)$$

$$+ \left[1 + \phi_T\left(D + 2D\sqrt{T}\right)\right]\sqrt{\log(\alpha_{0:T}/\alpha_0)}\sqrt{D^2(\alpha_0 + \max_t \alpha_t \log(\alpha_{0:T}/\alpha_0))}$$

$$+ \left[1 + \phi_T\left(D + 2D\sqrt{T}\right)\right]\sqrt{\sum_{t=1}^T \alpha_t\|v_t - \overline{v}_T\|^2} \tag{7}$$

Finally, observe that:

$$\left[1 + \phi_T\left(D + 2D\sqrt{T}\right)\right]\sqrt{\sum_{t=1}^T \alpha_t\|v_t - \overline{v}_T\|^2} + \phi_T(\|v_T - \mathring{u}\|)\sqrt{\alpha_0\|\overline{v}_T - \mathring{u}\|^2 + \sum_{t=1}^T \|\hat{g}_t\|^2\|\overline{v}_T - \mathring{u}\|^2}$$

$$\le \left[1 + 2\phi_T\left(D + 2D\sqrt{T}\right)\right]\sqrt{\alpha_0\|v_0 - \mathring{u}\|^2 + \sum_{t=1}^T \|\hat{g}_t\|^2\|v_t - \mathring{u}\|^2} \tag{8}$$

substituting (7) into (6), using (8) and recalling that $\alpha_t = \|\hat{g}_t\|^2$ for $t \ge 1$, we the desired bound.

$\square$

## 5 DISCUSSION AND CONNECTIONS WITH MOMENTUM

We have provided a method for incorporating arbitrary time-varying "guesses" for the comparison point $\mathring{u}$ into an online optimization algorithm. Our regret bound automatically improves as these guesses improve in quality. Our method is also relatively straightforward: simply add a (weighted) running average of the past guesses to the outputs of some base online optimization algorithm.

It will be instructive to re-write our algorithm in terms of "updates" rather than iterates. That is, let us define $\Delta_t$ by:

$$\hat{\Delta}_t = \hat{z}_{t+1} - \hat{z}_t$$

So, $\hat{\Delta}_t$ is the "update" that the base algorithm $\mathcal{A}$ is attempting to apply to the parameters $\hat{z}_t$. Our algorithm transforms this $\hat{\Delta}_t$ into a final $\Delta_t$:

$$\Delta_t = z_{t+1} - z_t$$

Let us compute $\Delta_t$ in terms of $\Delta_{t-1}$ in order to gain some more intuition for the mechanics of our algorithm. For simplicity, we will ignore the projections to the constraint set $W$. In this case, we have:

$$\Delta_t = \hat{z}_{t+1} + \overline{v}_t - \hat{z}_t - \overline{v}_{t-1}$$
$$= \hat{\Delta}_t + \frac{\alpha_t}{\alpha_{0:t}}(v_t - \overline{v}_{t-1})$$

Now, let us consider what this formula means for the natural choice $v_t = z_t$ employed by prior work. Let us also assume that $\alpha_t = \alpha_0$ for all $t$ in order to simplify the algebra. In this case we have $v_t - \overline{v}_{t-1} = \hat{z}_t = \hat{\Delta}_{1:t-1}$ and so we can define:

$$m_t = \frac{\hat{\Delta}_{1:t}}{t+1} = \left(1 - \frac{1}{t+1}\right) m_{t-1} + \frac{\hat{\Delta}_t}{t+1}$$
$$\Delta_t = \left(1 - \frac{1}{t+1}\right) \hat{\Delta}_t + \frac{m_t}{t+1}$$

That is, the update is equivalent to a form of momentum applied to the base update $\hat{\Delta}$ with a varying EMA value $\beta_t = \frac{t}{t+1}$. The formula for $\Delta_t$ is slightly different than the form for "Nesterov momentum" employed in typical machine learning optimization libraries Sutskever et al. (2013): $m_{t-1}$ is multiplied by $\beta_t$, when updating $m_t$, but by $1 - \beta_t$ when forming the final update.

Our development suggests several intriguing questions for further study. First, the MetaGrad algorithm of van Erven & Koolen (2016) achieves a more refined "full-matrix" guarantee than we present here:

$$R_T(\mathring{u}) \leq \tilde{O}\left(\sqrt{d \sum_{t=1}^{T} \langle g_t, z_t - u \rangle^2}\right)$$

Although this bound is incomparable to ours (due to the dimension factor), approximations of full-matrix algorithms (e.g. Shampoo) are empirically successful, suggesting that achieving this sort of bound is desirable. Our present technique is focused on the 2-norm, but it is plausible that a more advanced development would recover the full-matrix case.

Next, it is natural to wonder if it is possible to achieve both "primal" and "dual" optimism simultaneously, with a bound like:

$$R_T(\mathring{u}) \leq \sqrt{\sum_{t=1}^{T} \|g_t - h_t\|^2 \|v_t - \mathring{u}\|^2}$$

for any sequence of "dual hints" $h_t$ and "primal hints" $v_t$. Such a bound might have improved performance on strongly-convex and smooth objectives.

Finally, the focus of this paper is purely mathematical, and it remains to investigate the empirical properties of this method.

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

## A    APPENDIX

## B    PROOFS

Here is a technical result needed to prove our main Theorem:

**Theorem 3.** *Let $D = \sup_{x,y \in W} \|x - y\|$ and define $R_T^{\mathcal{A}}(u) = \sum_{t=1}^{T} \langle \hat{g}_t, \hat{z}_t - u \rangle$ be the regret of $\mathcal{A}$ on the gradients $\hat{g}_1, \ldots, \hat{g}_T$. Then for any vector $X$,*

$$\left\langle \sum_{t=1}^{T} \hat{g}_t, X \right\rangle \le \inf_k \frac{R_T^{\mathcal{A}}(-kX)}{k} + \frac{D\|\hat{g}_t\|_{1:T}}{k}$$

*Proof.* Let's consider the comparison point $\mathring{u} = -kX$. Then the regret of $\mathcal{A}$ is:

$$\sum_{t=1}^{T} \langle \hat{g}_t, \hat{z}_t \rangle + k \left\langle \sum_{t=1}^{T} \hat{g}_t, X \right\rangle = R_T^{\mathcal{A}}(-kX)$$

$$\left\langle \sum_{t=1}^{T} \hat{g}_t, X \right\rangle \le \frac{R_T^{\mathcal{A}}(-kX)}{k} - \frac{\sum_{t=1}^{T} \langle \hat{g}_t, \hat{z}_t \rangle}{k}$$

So to prove the Theorem we need only bound $\langle \hat{g}_t, \hat{z}_t \rangle$. We accomplish this using the properties of $\hat{g}_t$ described by (5). Specifically, since $\hat{g}_t$ is obtained by applying the constraint set reduction to $\hat{z}_t + \overline{v}_{t-1}$, we have for any $w \in W$:

$$\langle \hat{g}_t, \hat{z}_t \rangle = \langle \hat{g}_t, \hat{z}_t + \overline{v}_{t-1} - w \rangle + \langle \hat{g}_t, w - \overline{v}_{t-1} \rangle$$
$$\ge \langle g_t, z_t - w \rangle + \langle \hat{g}_t, w - \overline{v}_{t-1} \rangle$$

set $w = z_t \in W$:

$$\ge -D\|\hat{g}_t\|$$

$\square$

To see an example of this Theorem in action, consider an algorithm that obtains regret $R_T^{\mathcal{A}}(u) = \tilde{O}(\|u\|\sqrt{\sum_{t=1}^{t} \|\hat{g}_t\|^2})$. In this case, by taking $K$ sufficiently large, the Theorem shows that $\langle \hat{g}_{1:T}, X \rangle \le \tilde{O}(\|X\|\sqrt{\sum_{t=1}^{T} \|\hat{g}_t\|^2})$ for any $X$, and so $\|\hat{g}_{1:T}\| \le \tilde{O}(\sqrt{\sum_{t=1}^{T} \|g_t\|^2})$

Now we can restate our main Theorem:

**Theorem 2.** *Suppose $\mathcal{A}$ guarantees regret $R_t^{\mathcal{A}}(u) = \sum_{i=1}^{t} \langle \hat{g}_i, \hat{z}_i - u \rangle \le \epsilon + \psi_t(\|u\|)\|u\| + \phi_t(\|u\|)\|u\|\sqrt{\alpha_0 + \sum_{i=1}^{t} \|\hat{g}_i\|^2}$ for all $u$ and $t$, where $\epsilon > 0$, and $\psi_t(x)$ and $\phi_t(x)$ are increasing functiosn of $t$ and $x$. Then, Algorithm 1 guarantees both $\|\hat{g}_t\| \le \|g_t\|$ for all $t$ and also:*

$$\sum_{t=1}^{T} \langle g_t, z_t - \mathring{u} \rangle \le \epsilon + \|v_T - \mathring{u}\|\psi_T(\|v_T - \mathring{u}\|) + \left[\epsilon + D\psi_T\left(D + 2D\sqrt{T}\right)\right]\log(1 + \|\hat{g}\|_{1:T}^2/\alpha_0)$$

$$+ \left[1 + \phi_T\left(D + 2D\sqrt{T}\right)\right]\sqrt{\log(1 + \|\hat{g}\|_{1:T}^2/\alpha_0)}\sqrt{D^2(\alpha_0 + \max_t \|\hat{g}_t\|^2 \log(1 + \|\hat{g}\|_{1:T}^2/\alpha_0))}$$

$$+ \left[1 + 2\phi_T\left(D + 2D\sqrt{T}\right)\right]\sqrt{\alpha_0\|v_0 - \mathring{u}\|^2 + \sum_{t=1}^{T} \|\hat{g}_t\|^2\|v_t - \mathring{u}\|^2}$$

*Proof.* Define $\alpha_t = \|g_t\|^2$ for $t \ge 1$. Then following Cutkosky & Orabona (2018); Cutkosky & Boahen (2017), we consider for purposes of intuition a random variable $V$ that takes on value $v_t$

with probability proportional to $\alpha_t$. By definition we have

$$\mathbb{E}[V] = \frac{\sum_{t=0}^{T} \alpha_t v_t}{\sum_{t=0}^{T} \alpha_t} = \overline{v}_T$$

$$\sum_{t=0}^{T} \alpha_t \mathbb{E}[\|V - \mathring{u}\|^2] = \sum_{t=0}^{T} \alpha_t \|v_t - \mathring{u}\|^2$$

and by bias-variance decomposition we can further write

$$\sum_{t=1}^{T} \alpha_t (\mathbb{E}[\|V - \mathbb{E}[V]\|^2] + \|\mathbb{E}[V] - \mathring{u}\|^2) = \sum_{t=1}^{T} \alpha_t \|v_t - \mathring{u}\|^2$$

so that we can rephrase our objective as (ignoring constants, logarithmic terms, and $\epsilon$, $\psi$ and $\phi$ dependencies):

$$R_T(\mathring{u}) \leq \epsilon + 2\sqrt{\sum_{t=0}^{T} \alpha_t (\mathbb{E}[\|V - \mathbb{E}[V]\|^2] + \|\mathbb{E}[V] - \mathring{u}\|^2)}$$

Since $\sqrt{a} + \sqrt{b} \leq 2\sqrt{a+b}$, it suffices to show

$$R_T(\mathring{u}) \leq \epsilon + \sqrt{\sum_{t=0}^{T} \alpha_t \|\mathbb{E}[V] - \mathring{u}\|^2} + \sqrt{\sum_{t=1}^{T} \alpha_t \mathbb{E}[\|V - \mathbb{E}[V]\|^2]}$$

$$= \epsilon + \sqrt{\sum_{t=0}^{T} \alpha_t \|\overline{v}_T - \mathring{u}\|^2} + \sqrt{\sum_{t=0}^{T} \alpha_t \|v_t - \overline{v}_T\|^2}$$

Keeping this intuition in mind, let us examine the regret:

$$R_T(\mathring{u}) = \sum_{t=1}^{T} \langle g_t, z_t - \mathring{u} \rangle$$

using (5):

$$\leq \sum_{t=1}^{T} \langle \hat{g}_t, \hat{z}_t^v - \mathring{u} \rangle$$

$$= \sum_{t=1}^{T} \langle \hat{g}_t, \hat{z}_t + \overline{v}_{t-1} - \mathring{u} \rangle$$

$$= \sum_{t=1}^{T} \langle \hat{g}_t, \hat{z}_t - (\mathring{u} - \overline{v}_T) \rangle + \sum_{t=1}^{T} \langle \hat{g}_t, \overline{v}_{t-1} - \overline{v}_T \rangle$$

$$= R_T^A(\mathring{u} - \overline{v}_T) + \sum_{t=1}^{T} \langle \hat{g}_t, \overline{v}_{t-1} - \overline{v}_T \rangle$$

$$\leq \epsilon + \|v_T - \mathring{u}\| \psi_T(\|v_T - \mathring{u}\|) + \phi_T(\|v_T - \mathring{u}\|)\sqrt{\alpha_0\|v_T - \mathring{u}\|^2 + \sum_{t=1}^{T} \|g_t\|^2 \|v_T - \mathring{u}\|^2} + \sum_{t=1}^{T} \langle \hat{g}_t, \overline{v}_{t-1} - \overline{v}_T \rangle$$

The first-few terms in the above expression roughly match the "bias" part of the bias-variance decomposition. So, it remains to prove that (again ignoring constants and logarithmic terms):

$$\sum_{t=1}^{T} \langle \hat{g}_t, \overline{v}_{t-1} - \overline{v}_T \rangle \leq \sqrt{\sum_{t=0}^{T} \alpha_t \|v_t - \overline{v}_T\|^2}$$

Let us start by calculating an alternative form for this sum:

$$\sum_{t=1}^{T}\langle \hat{g}_t, \overline{v}_{t-1} - \overline{v}_T \rangle - \sum_{t=1}^{T-1}\langle \hat{g}_t, \overline{v}_{t-1} - \overline{v}_{T-1} \rangle = \left\langle \sum_{t=1}^{T} \hat{g}_t, \overline{v}_{T-1} - \overline{v}_T \right\rangle$$

So that telescoping we have

$$\sum_{t=1}^{T}\langle \hat{g}_t, \overline{v}_{t-1} - \overline{v}_T \rangle = \sum_{t=1}^{T}\langle \hat{g}_{1:t}, \overline{v}_{t-1} - \overline{v}_t \rangle$$

Now we apply Lemma 3 to see that for some $K_t$ to be chosen later:

$$\langle \hat{g}_{1:t}, \overline{v}_{t-1} - \overline{v}_t \rangle \leq \left[ \frac{R_t^{\mathcal{A}}\left(K \frac{\overline{v}_{t-1} - \overline{v}_t}{\|\overline{v}_{t-1} - \overline{v}_t\|}\right)}{K_t} + \frac{D \sum_{i=1}^{t}\|\hat{g}_i\|}{K_t} \right] \|\overline{v}_{t-1} - \overline{v}_t\|$$

for any $K_t$. Use the assumed form of $R_t^{\mathcal{A}}$ to obtain:

$$\langle \hat{g}_{1:t}, \overline{v}_{t-1} - \overline{v}_t \rangle \leq \left[ \frac{\epsilon + \sum_{i=1}^{t}\|\hat{g}_i\|D}{K_t} + \psi_t(K) + \phi_t(K_t)\sqrt{\sum_{i=1}^{t}\|\hat{g}_i\|^2} \right] \|\overline{v}_{t-1} - \overline{v}_t\|$$

Next, we bound $\|\overline{v}_{t-1} - \overline{v}_t\|$:

$$\alpha_{0:t}\overline{v}_t - \alpha_{0:t-1}\overline{v}_{t-1} = \alpha_t v_t$$

$$\overline{v}_t - \overline{v}_{t-1} = \frac{\alpha_t(v_t - \overline{v}_{t-1})}{\alpha_{0:t}}$$

So that

$$\langle \hat{g}_{1:t}, \overline{v}_{t-1} - \overline{v}_t \rangle \leq \frac{(\epsilon + \sum_{i=1}^{t}\|\hat{g}_i\|D)\|v_t - \overline{v}_{t-1}\|\alpha_t}{K_t \alpha_{0:t}} + \psi_t(K_t)\frac{\alpha_t\|v_t - \overline{v}_{t-1}\|}{\alpha_{0:t}} + \phi_t(K_t)\frac{\alpha_t\|v_t - \overline{v}_{t-1}\|}{\sqrt{\alpha_{0:t}}}$$

Now, we set $K_t$ to be:

$$K_t = \frac{\epsilon + \sum_{i=1}^{t}\|\hat{g}_i\|D}{\epsilon/D + \sqrt{\alpha_{0:t}}} \leq D(1 + 2\sqrt{T})$$

The inequality follows since $\sum_{i=1}^{t}\|\hat{g}_i\| \leq \sqrt{T}\sqrt{\sum_{i=1}^{t}\|\hat{g}_i\|^2} \leq \sqrt{T}\sqrt{\alpha_{0:t}}$.

With this value, we have:

$$\langle \hat{g}_{1:t}, \overline{v}_{t-1} - \overline{v}_t \rangle \leq \epsilon \cdot \frac{\alpha_t}{\alpha_{0:t}} + \psi_t(K_t)\frac{\alpha_t\|v_t - \overline{v}_{t-1}\|}{\alpha_{0:t}} + (1 + \phi_t(K_t))\frac{\alpha_t\|v_t - \overline{v}_{t-1}\|}{\sqrt{\alpha_{0:t}}}$$

Let's define $K = \max_t K_t \leq D(1 + 2\sqrt{T})$. Then, since $\psi_t$ and $\phi_t$ are increasing in $t$, we have:

$$\sum_{t=1}^{T}\langle \hat{g}_{1:t}, \overline{v}_{t-1} - \overline{v}_t \rangle \leq (\epsilon + D\psi_T(K))\log(\alpha_{0:T}/\alpha_0) + (1 + \phi_T(K))\sqrt{\log(\alpha_{0:T}/\alpha_0)}\sqrt{\sum_{t=1}^{T}\alpha_t\|v_t - \overline{v}_{t-1}\|^2}$$

Now observe that $\overline{v}_{t-1}$ is actually the output of the Follow-the-Leader algorithm on the losses $f_t(\overline{v}) = \alpha_t\|v_t - \overline{v}\|^2$. Therefore by standard analysis of this algorithm (e.g. see McMahan (2014)), we have $\sum_{t=1}^{T}\alpha_t\|v_t - \overline{v}_{t-1}\|^2 - \alpha_t\|v_t - \overline{v}_T\|^2 \leq 2\max_t \alpha_t D^2(1 + \log(\alpha_{0:T}/\alpha_0))$, so that we have

$$\sum_{t=1}^{T} \langle \hat{g}_{1:t}, \overline{v}_{t-1} - \overline{v}_t \rangle \le (\epsilon + D\psi_T(K)) \log(\alpha_{0:T}/\alpha_0)$$

$$+ (1 + \phi_T(K))\sqrt{\log(\alpha_{0:T}/\alpha_0)} \sqrt{2 \max_t \alpha_t D^2(1 + \log(\alpha_{0:T}/\alpha_0)) + \sum_{t=1}^{T} \alpha_t \|v_t - \overline{v}_T\|^2}$$

$$\le \left[ \epsilon + D\psi_T \left( D + 2D\sqrt{T} \right) \right] \log(\alpha_{0:T}/\alpha_0)$$

$$+ \left[ 1 + \phi_T \left( D + 2D\sqrt{T} \right) \right] \sqrt{\log(\alpha_{0:T}/\alpha_0)} \sqrt{2 \max_t \alpha_t D^2(1 + \log(\alpha_{0:T}/\alpha_0))}$$

$$+ \left[ 1 + \phi_T \left( D + 2D\sqrt{T} \right) \right] \sqrt{\sum_{t=1}^{T} \alpha_t \|v_t - \overline{v}_T\|^2}$$

So, overall we have a regret bound:

$$R_T(\mathring{u}) \le \epsilon + \|v_T - \mathring{u}\|\psi_T(\|v_T - \mathring{u}\|) + \phi_T(\|v_T - \mathring{u}\|)\sqrt{\alpha_0\|v_T - \mathring{u}\|^2 + \sum_{t=1}^{T} \|g_t\|^2 \|v_T - \mathring{u}\|^2}$$

$$+ \left[ \epsilon + D\psi_T \left( D + 2D\sqrt{T} \right) \right] \log(\alpha_{0:T}/\alpha_0)$$

$$+ \left[ 1 + \phi_T \left( D + 2D\sqrt{T} \right) \right] \sqrt{\log(\alpha_{0:T}/\alpha_0)} \sqrt{2 \max_t \alpha_t D^2(1 + \log(\alpha_{0:T}/\alpha_0))}$$

$$+ \left[ 1 + \phi_T \left( D + 2D\sqrt{T} \right) \right] \sqrt{\sum_{t=1}^{T} \|\hat{g}_t\|^2 \|v_t - \overline{v}_T\|^2}$$

$$\le \epsilon + \|v_T - \mathring{u}\|\psi_T(\|v_T - \mathring{u}\|) + \left[ \epsilon + D\psi_T \left( D + 2D\sqrt{T} \right) \right] \log(1 + \|\hat{g}\|_{1:T}^2/\alpha_0)$$

$$+ \left[ 1 + \phi_T \left( D + 2D\sqrt{T} \right) \right] \sqrt{\log(1 + \|\hat{g}\|_{1:T}^2/\alpha_0)} \sqrt{2 \max_t \|\hat{g}_t\|^2 D^2(1 + \log(1 + \|\hat{g}\|_{1:T}^2/\alpha_0))}$$

$$+ \left[ 1 + 2\phi_T \left( D + 2D\sqrt{T} \right) \right] \sqrt{\alpha_0\|v_0 - \mathring{u}\|^2 + \sum_{t=1}^{T} \|\hat{g}_t\|^2 \|v_t - \mathring{u}\|^2}$$

$\square$