# OpenReview forum: "Primal Optimism in Online Optimization"
_ICLR.cc/2026/Conference — Submitted to ICLR 2026_

### Official Review · Reviewer_3yMY · 2025-10-20

**Soundness:** 4
**Presentation:** 3
**Contribution:** 2
**Rating:** 4
**Confidence:** 3

**Summary:**

The paper presents an approach for low regret when the optimal point for an online convex optimisation can be accurately predicted.  The main contribution is the theoretical worst-case analysis of this algorithm.

**Strengths:**

The paper is well written, and as far as I can tell (I did not check every part of the maths) the analysis is sound.

**Weaknesses:**

The justification by reference to ML training made in the introduction is largely spurious - neural net training is non-convex, so none of the analysis presented applies.   The focus on worst-case analysis also weakens the contribution.  More generally, analysis of online convex learning algorithms is a mature area and the present paper feels like only a relatively minor addition to that literature, in my view not really enough for a flagship conference like ICLR.

**Questions:**

See comments re weaknesses above.

---

> ### Author Response · Authors · 2025-11-17
> **response to review**
>
> Thanks for you comments. We’d like to clarify a couple points:
>
> * Focus on worst-case analysis: actually our algorithm is an adaptive algorithm, so it specifically focuses on beyond-worst-case analysis. Perhaps by “worst-case”, you are objecting to the non-stochastic nature of online optimization. However, it turns out (surprisingly) that this adversarial setting does not harm stochastic results: almost all guarantees achievable by dropping it (e.g. bounds depending on variance of gradient etc) are easy corollaries of adaptive online optimization analysis. See e.g. our references [Cutkosky 2019], [Kavis et al 2019], or also [Joulani et al 2020]. In fact, usually the proof and algorithm is much simpler using the online optimization analysis. We think this is a large part of the reason for the popularity of online optimization work.
>
> * Spurious motivation: our primary contribution is mathematical, and of course it is true that neural networks are not convex. We are happy to include any general other motivations for the study of online optimization that you might find in various textbooks; online optimization is certainly a field of general interest. However, in defense of our initial motivation, it is a fact that online optimism has an exceptional track record of inspiring practical neural network optimizers (since most algorithms in widespread practical use today trace their roots to AdaGrad and Shampoo). Understanding why this might be is its own mystery.

---

### Official Review · Reviewer_QGfz · 2025-10-30

**Soundness:** 3
**Presentation:** 4
**Contribution:** 3
**Rating:** 8
**Confidence:** 3

**Summary:**

This paper presents an algorithm for online convex optimization with hints for the decision vector. This differs from most existing work that output hints for the gradient, i.e. optimism. This develops an algorithm for this setting that has adaptive regret bounds in the sense that the regret is bounded by $\sqrt{\sum_{t=1}^T || g_t ||^2 || v_t - u ||}$, where $g_t$ are the cost vectors, $v_t$ are the hints for the decision vector, and $u$ is the comparator.

**Strengths:**

1. It appears to be a new approach to consider hints for the decision vector ("primal optimism"). This problem seems to be a nice counterpart to hints on the gradients. Furthermore, it is well-motivated by its relevance for the analysis of strongly-convex stochastic optimization.
2. The paper presents a very clear exposition of the proposed approach and its analysis. I found it insightful and quite easy to follow.
3. The paper clearly discusses how the work relates to existing approaches in the online optimization and stochastic optimization literatures.

**Weaknesses:**

1. The approach appears to be a fairly straightforward application of existing techniques, although I do think that it is valuable to bring these techniques together for a new problem setting.

**Questions:**

1. In the first line of the abstract, it is not quite clear to say: "classic online convex optimization problem in which an algorithm outputs vectors $z_t$ in response to vectors $g_t$." Its somewhat confusing because the $g_t$ are not revealed until after $z_t$ are chosen so its not really "in response to".
2. On line 122, there is the notation $H^\star$, which is not defined. It seems that it's supposed to be $H$.

---

> ### Author Response · Authors · 2025-11-17
> **response to review**
>
> Thank you very much for your comments.
>
> We appreciate your comments on the exposition; we worked hard to make the setting and methods clear before going into technical details. We agree that our primary contribution is the introduction of the new type of regret bound, which we hope will inspire further applications beyond the ones we have already provided.
>
> We will address the issues you pointed out in the questions (indeed, $H^\star$ should be $H$).

---

### Official Review · Reviewer_PY9x · 2025-10-31

**Soundness:** 2
**Presentation:** 2
**Contribution:** 2
**Rating:** 4
**Confidence:** 3

**Summary:**

The paper studies the topic of online convex optimization and theoretically analyzes the algorithm's regret in this setting. They consider a black-box unconstrained optimizer with known regret bounds and utilize that to achieve primal optimistic regret guarantees, where hints are given for the optimal parameter and are possibly changing with time.

**Strengths:**

Originality:
The paper studies optimistic optimization from a new perspective.

Quality:
Many parts of the submission appear technically correct.

Clarity:
The general structure of the text is clear.

Significance:
Theoretical novel findings are present in the form of regret guarantees, which are improved with respect to the existing literature.

**Weaknesses:**

I am leaning towards rejection. Below are my reasons.

1. Experiments are missing as a whole.
2. The results are promising. However, the challenges involved are not clear. The direction taken seems rather straightforward in this field of research. Please clarify exactly why primal optimism has not been studied like this before. Is it a motivation issue? If not, what is the exact novelty in your approach that helped you achieve primal optimism unlike the others before.

3. Certain parts of the paper have skipped some explanations, resulting in a lack of rigor, see "Questions" for examples.
4. A substantial number of writing mistakes (such as typos) are present, see "Questions" for examples.

**Questions:**

Questions:

Page 6 Corollary 1: it is not justified how the simple $D$ terms are eliminated in the guarantee as opposed to Theorem 2. How exactly?

Page 7 Line 327: where did this objective come from?

Page 7 Line 371: should $K$ be $K_t$ instead? and the argument of regret in RHS should be additive inverse?

Page 8 Equation 8: it is not clear how (8) is used. How exactly?


Suggestions:

Page 3 Line 125: clarify if $v_t$ is revealed before $z_t$ or $z_{t+1}$.

Page 6 Line 323: sum should start at index $0$.

Page 8 Line 400: explicitly explain how the log terms are generated.


Minor Comments:

Page 3 Line 110: correct the grammar in this paragraph.

Page 6 Algorithm 1: correct typos, e.g., $v_9$ in Line 272.

Page 6 Theorem 2: correct typos.

---

> ### Author Response · Authors · 2025-11-17
> **response to review**
>
> Thanks very much for your careful reading and comments on the analysis. After our general response we include answers to your questions about the proof. We will of course also address the typos you identified.
>
> * Why this property was not previously obtained: frankly, we feel that primal optimism is one of those formulations that is “obvious in hindsight”, but not in foresight. In particular, our motivating examples fit well within the common interests of the online learning community (automatic adaptivity to certain problem parameters), and the need to consider these more advanced online-to-batch conversions was present as far back as 2019. It is true that recent work has brought these conversions more publicity however.
>
> * Regarding the difficulty of our proofs: We actually agree that, once one has realized that primal optimism is a good bound to obtain, then our actual proof is not extremely complicated (although again, hindsight is always 20/20). Frankly many proofs in online optimization are not too complicated: even the proof of the original “dual” optimism is fairly straightforward. In this sense, our contribution is more a careful framing of a useful and obtainable algorithmic property, and an initial argument for how to obtain it. Just like with dual optimism, we expect there more diverse approaches yet to be discovered.
>
>
>
> ### Technical Questions about the proof:
>
> * Simple D terms in Corollary: The hypothesis of the corollary suggests that $\phi$ and $\psi$ are logarithmic, so those functions are absorbed into the O-tilde. The only other “simple D term” that we can see in Theorem 2 is multiplied by a logarithmic term rather than a $\sqrt{\sum_{t=1}^T \|g_t\|^2}$ term, so we also placed it in the O-\tilde.
>
> * Rephrasing objective: by the previous line, this expression would imply the desired equation (4). Perhaps it would be more proper to include a O-\tilde around the expression: this is meant as an aspirational guide for how the proof will proceed rather than a concrete statement.
>
> * Line 371: yes you are correct.
>
> * Equation (8): This equation observes that we can bound the $\phi$ term that does not have constant argument present in equation 6 by one that does have a constant argument and group it together with a similar term in equation (7).

---

### Official Review · Reviewer_4gNt · 2025-11-05

**Soundness:** 2
**Presentation:** 2
**Contribution:** 2
**Rating:** 4
**Confidence:** 3

**Summary:**

This paper studies the online convex optimization problem and proposes methods that incorporate optimism with respect to the comparator. Based on existing parameter-free online learning methods, the paper provides a reduction to achieve regret bounds that can be tight when the gap between the comparator and the optimistic sequence is small.

**Strengths:**

- The idea of introducing optimism in the primal space is interesting. The authors demonstrate its usefulness in stochastic optimization with strongly convex functions, showing that the proposed method can adapt to both convex and Lipschitz as well as strongly convex and Lipschitz settings.

- The proposed approach benefits from a gradient-based update and avoids the need to run multiple algorithms in parallel, as required by methods such as MetaGrad (Van Erven & Koolen, 2016) and Müller’s algorithm (Wang et al., 2020).

**Weaknesses:**

- My main concern lies in the technical novelty of the paper. The proposed method and analysis (e.g., the decomposition in Section 3.1) appear quite similar to Algorithm 6 in Cutkosky and Orabona (2018). The primary difference seems to be that the paper chooses $v_t$ as the learner’s prediction and generalizes it to a more flexible form. However, it is not clear how challenging this generalization actually is, or how much additional insight it brings beyond the previous work.


- Hidden logarithmic factors: The use of big-$O$ and $\leq$ notation obscures logarithmic dependencies, which makes comparisons with prior work potentially misleading. For example, in stochastic strongly convex optimization, the optimal convergence rate is $O(1/T)$ without logarithmic factors (Cutkosky 2019). However, since the proposed method builds upon a parameter-free base algorithm, additional logarithmic factors are introduced, resulting in a rate that is not optimal. I strongly recommend that the authors make the dependence on $\log T$ explicit in the paper.


- Inaccurate claims and unclear statements:
  - Line 58: In Rakhlin and Sridharan (2013), the proposed bound does not adapt to the comparator; rather, it scales with the diameter of the comparator space.
  - Line 86: In Van Erven & Koolen (2016), when $v_t = z_t$, the regret bound they obtain is actually stronger than the one in this paper, as it takes the form $\sqrt{\sum\_{t=1}^T (g_t^\top (z_t - u))^2}.$

**Questions:**

- Could the authors clearly highlight the main technical challenges and contributions beyond Cutkosky and Orabona (2018).

- What is the significance of introducing the primal hint? In particular, for stochastic strongly convex optimization, the obtained results are not optimal up to logarithmic factors in  $T$. It appears difficult to eliminate these $\log T$ terms due to the reliance on a parameter-free base algorithm.

---

> ### Author Response · Authors · 2025-11-17
> **response to review**
>
> Thanks very much for your review. Below we answer your questions:
>
> Questions:
>
> The main contribution of this paper is to open up investigation into primal optimism in regret bounds. In particular, it is NOT to provide the most technically challenging result possible.
>
> We agree that *once one has identified primal optimism as desirable*, then our proof is not too challenging for someone familiar with the relevant techniques. We don’t see this as a significant flaw: indeed many classical proofs in online convex optimization are actually extremely easy (far easier than the present analysis): the key is articulating the desirable regret properties. Arguably, it is in fact a feature.
>
> Significance of primal hints: In the application we discussed, the convergence rate is indeed suboptimal by logarithmic factors given our current method of achieving primal optimism, but it does have other desirable properties: the bound automatically depends on the “correct” strong convexity parameter. In fact, a more refined analysis would show that the bound can depend on a different (unknown) strong-convexity parameter for each iteration. We aren’t aware of any other way to achieve such a result in the context of these more advanced online-to-batch conversions, even at the expense of our logarithmic factors. For example, this enables a bound that has a simultaneously a last-iterate guarantee while also not requiring knowledge of strong-convexity parameters.
>
> Regarding Inaccurate claims:
> We attempted to optimize for clearly communicating the goals by providing the most direct-possible comparison to prior work without excessive details, but you are of course correct that Rakhlin & Sridharan require one to tune the algorithm using an unknown value of ||u|| to obtain the stated result, and that van Ervan & Koolen achieves a slightly different guarantee which we discuss in more detail at the end of the paper. We will make this part clearer. Note that providing these details up-front only serves to further differentiate our results from prior literature.

---

### Meta-Review · Area_Chair_bfzq · 2026-01-12

**Summary:**

1) Most reviewers have concerns on the technical novelty of the paper, where the direction taken seems rather straightforward in this field of research;

2) Inaccurate claims and unclear statements, and a substantial number of writing mistakes;

3) Experiments are missing as a whole;

4) The justification by reference to ML training made in the motivation is largely spurious.

**Reviewer Concerns:**

1) The authors argue that although the proof is not too challenging for someone familiar with the relevant techniques, the key is articulating the desirable regret properties, which may be a feature. However, both the technical challenge and the motivation why primal optimism has not been studied like this before are not been addressed;

2) We recommend the authors to further modify the paper writting;

3) No additional experiments are provided;

4) Mathematical contributions are well suite ICLR, but the reviewers think the theoretical part are not very novel.

**Reviewer Scores:**

No

---

### Decision · Program_Chairs · 2026-01-26

Reject